# Enabling Limited Resource-Bounded Disjunction in Scheduling

**Jagriti Agrawal, Wayne Chi, Steve Chien, Gregg Rabideau, Stephen Kuhn, and Dan Gaines**

Jet Propulsion Laboratory
California Institute of Technology
4800 Oak Grove Drive
Pasadena, CA 91109
{firstname.lastname}@jpl.nasa.gov

## Abstract

We describe three approaches to enabling an extremely computationally limited embedded scheduler to consider a small number of alternative activities based on resource availability. We consider the case where the scheduler is so computationally limited that it cannot backtrack search. The first two approaches precompile resource checks (called guards) that only enable selection of a preferred alternative activity if sufficient resources are estimated to be available to schedule the remaining activities. The final approach mimics backtracking by invoking the scheduler multiple times with the alternative activities. We present an evaluation of these techniques on mission scenarios (called sol types) from NASA's next planetary rover where these techniques are being evaluated for inclusion in an onboard scheduler.

## Introduction

Embedded schedulers must often operate with very limited computational resources. Due to such limitations, it is not always feasible to develop a scheduler with a backtracking search algorithm. This makes it challenging to perform even simple schedule optimization when doing so may use resources needed for yet unscheduled activities.

In this paper, we present three algorithms to enable such a scheduler to consider a very limited type of preferred activity while still scheduling all required (hereafter called *mandatory*) activities. Preferred activities are grouped into *switch groups*, sets of activities, where each activity in the set is called a *switch case*, and exactly one of the activities in the set must be scheduled. They differ only by how much time, energy, and data volume they consume and the goal is for the scheduler to schedule the most desirable activity (coincidentally the most resource consuming activity) without sacrificing any other mandatory activity.

The target scheduler is a non-backtracking scheduler to be onboard the NASA Mars 2020 planetary rover (Rabideau and Benowitz 2017) that schedules in priority first order and never removes or moves an activity after it is placed during a single run of the scheduler. Because the scheduler does not backtrack, it is challenging to ensure that scheduling a consumptive switch case will not use too many resources

and therefore prevent a later (in terms of scheduling order, not necessarily time order) mandatory activity from being scheduled.

The onboard scheduler is designed to make the rover more robust to run-time variations by rescheduling multiple times during execution (Gaines et al. 2016a). If an activity ends earlier or later than expected, then rescheduling will allow the scheduler to consider changes in resource consumption and reschedule accordingly. Our algorithms to schedule switch groups must also be robust to varying execution durations and rescheduling.

We have developed several approaches to handle scheduling switch groups. The first two, called guards, involve reserving enough sensitive resources (time, energy, data volume) to ensure all later required activities can be scheduled. The third approach emulates backtracking under certain conditions by reinvoking the scheduler multiple times. These three techniques are currently being considered for implementation in the Mars 2020 onboard scheduler.

## Problem Definition

For the scheduling problem we adopt the definitions in (Rabideau and Benowitz 2017). The scheduler is given

- a list of activities
  $A_1 \langle p_1, d_1, R_1, e_1, dv_1, \Gamma_1, T_1, D_1 \rangle \dots$
  $A_n \langle p_n, d_n, R_n, e_n, dv_n, \Gamma_n, T_n, D_n \rangle$

- where $p_i$ is the scheduling priority of activity $A_i$;

- $d_i$ is the nominal, or predicted, duration of activity $A_i$;

- $R_i$ is the set of unit resources $R_{i_1} \dots R_{i_m}$ that activity $A_i$ will use;

- $e_i$ and $dv_i$ are the rates at which the consumable resources energy and data volume respectively are consumed by activity $A_i$;

- $\Gamma_{i_1} \dots \Gamma_{i_r}$ are non-depletable resources used such as sequence engines available or peak power for activity $A_i$;

- $T_i$ is a set of start time windows $[T_{i_{j\_start}}, T_{i_{j\_preferred}}, T_{i_{j\_end}}] \dots [T_{i_{k\_start}}, T_{i_{k\_preferred}}, T_{i_{k\_end}}]$ for activity $A_i$. [1] ;

---

[1] If a preferred start time, $T_{i_{j\_preferred}}$ is not specified for window $j$ then it is by default $T_{i_{j\_start}}$

- $D_i$ is a set of activity dependency constraints for activity $A_i$ where $A_p \rightarrow A_q$ means $A_q$ must execute successfully before $A_p$ starts.

The goal of the scheduler is to schedule all mandatory activities and the best switch cases possible while respecting individual and plan-wide constraints.

Each activity is assigned a *scheduling priority*. This priority determines the order in which the activity will be considered for addition to the schedule. The scheduler attempts to schedule the activities in priority order, therefore: (1) higher priority activities can block lower priority activities from being scheduled and (2) higher priority activities are more likely to appear in the schedule.

*Mandatory Activities* are activities, $m_1 \dots m_j \subseteq A$, that must be scheduled. The presumption is that the problem as specified is *valid*, that is to say that a schedule exists that includes all of the mandatory activities, respects all of the provided constraints, and does not exceed available resources.

In addition, activities can be grouped into *Switch Groups*. The activities within a switch group are called *switch cases* and vary by how many resources (time, energy, and data volume) they consume. It is mandatory to schedule exactly one switch case and preferable to schedule a more resource intensive one, but not at the expense of another mandatory activity. For example, one of the Mars 2020 instruments takes images to fill mosaics which can vary in size; for instance we might consider $1x4$, $2x4$, or $4x4$ mosaics. Taking larger mosaics might be preferable, but taking a larger mosaic takes more time, takes more energy, and produces more data volume. These alternatives would be modeled by a switch group that might be as follows:

$$SwitchGroup = \begin{cases} Mosaic_{1x4} & d = 100 \text{ sec} \\ Mosaic_{2x4} & d = 200 \text{ sec} \\ Mosaic_{4x4} & d = 400 \text{ sec} \end{cases} \quad (1)$$

The desire is for the scheduler to schedule the activity $Mosaic_{4x4}$ but if it does not fit then try scheduling $Mosaic_{2x4}$, and eventually try $Mosaic_{1x4}$ if the other two fail to schedule. It is not worth scheduling a more consumptive switch case if doing so will prevent a future, lower priority mandatory activity from being scheduled due to lack of resources. Because our computationally limited scheduler cannot search or backtrack, it is a challenge to predict if a higher level switch case will be able to fit in the schedule without consuming resources that will cause another lower priority mandatory activity to be forced out of the schedule.

Consider the following example in Figure 1 where the switch group consists of activities B1, B2, and B3 and $d_{B3} > d_{B2} > d_{B1}$. Each activity in this example also has one start time window from $T_{i_{start}}$ to $T_{i_{end}}$.

B3 is the most resource intensive and has the highest priority so the scheduler will first try scheduling B3. As shown in Figure 1a, scheduling B3 will prevent the scheduler from placing activity C at a time satisfying its execution constraints. So, B3 should not be scheduled.

The question might arise as to why switch groups cannot simply be scheduled last in terms of scheduling order. This is difficult for several reasons: 1) We would like to avoid gaps

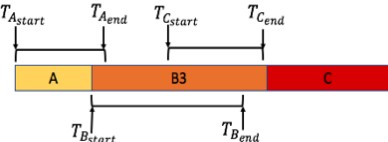

(a) Scheduling B3 first prevents activity C from being scheduled within its start time window.

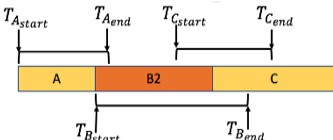

(b) B2 can be successfully scheduled without dropping any other mandatory activities.

Figure 1: Challenge to Schedule Switch Cases.

in the schedule which is most effectively done by scheduling primarily left to right temporally, and 2) if another activity is dependent on an activity in a switch group, then scheduling the switch group last would introduce complications to ensure that the dependencies are satisfied.

The remainder of the paper is organized as follows. First, we describe several plan wide energy constraints that must be satisfied. Then, we discuss two guard approaches to schedule preferred activities, which place conditions on the scheduler that restrict the placement of switch cases under certain conditions. We then discuss various versions of an approach which emulates backtracking by reinvoking the scheduler multiple times with the switch cases. We present empirical results to evaluate and compare these approaches.

## Energy Constraints

There are several energy constraints which must be satisfied throughout scheduling and execution. The scheduling process for each *sol*, or Mars day, begins with the assumption that the rover is asleep for the entire time spanning the sol. Each time the scheduler places an activity, the rover must be awake so the energy level declines. When the rover is asleep the energy level increases.

Two crucial energy values which must be taken into account are the *Minimum State of Charge (SOC)* and the *Minimum Handover State of Charge*. The state of charge, or energy value, cannot dip below the Minimum SOC at any point. If scheduling an activity would cause the energy value to dip below the Minimum SOC, then that activity will not be scheduled. In addition, the state of charge cannot be below the *Minimum Handover SOC* at the *Handover Time*, in effect when the next schedule starts (e.g., the handover SOC of the previous plan is the expected beginning SOC for the subsequent schedule).

In order to preserve battery life, the scheduler must also consider the *Maximum State of Charge* constraint. Exceeding the Maximum SOC hurts long term battery performance and the rover will perform *shunting*. To prevent it from exceeding this value, the rover may be kept awake.

## Guard Approaches

First we will discuss two guard methods to schedule switch cases, the Fixed Point guard and the Sol Wide guard. Both of these methods attempt to schedule switch cases by reserving enough time and energy to schedule the remaining mandatory activities. For switch groups, this means that resources will be reserved for the least resource consuming activity since it is mandatory to schedule exactly one activity in the switch group. The method through which both of these guard approaches reserve enough time to schedule future mandatory activities is the same. They differ in how they ensure there is enough energy. While the Fixed Point guard reserves enough energy at a single fixed time point - the time at which the least resource consuming switch case is scheduled to end in the nominal schedule, the Sol Wide guard attempts to reserve sufficient energy by keeping track of the energy balance in the entire plan, or sol.

In this discussion, we do not attempt to reserve data volume while computing the guards as it is not expected to be as constraining of a resource as time or energy. We aim to take data volume into account as we continue to do work on this topic.

Both the time and energy guards are calculated offline before execution occurs using a nominal schedule. Then, while rescheduling during execution, the constraints given by the guards are applied to ensure that scheduling a higher level switch case will not prevent a future mandatory activity from being scheduled. If activities have ended sufficiently early and freed up resources, then it may be possible to reschedule with a more consumptive switch case.

### Guarding for Time

First, we will discuss how the Fixed Point and Sol Wide guards ensure enough time will be reserved to schedule remaining mandatory activities while attempting to schedule a more resource consuming switch case.

If a preferred time, $T_{i_{j\_preferred}}$, is specified for an activity, the scheduler will try to place an activity closest to its preferred time while obeying all other constraints. Otherwise, the scheduler will try to place the activity as early as possible.

Each switch group in the set of activities used to create a *nominal schedule* includes only the nominal, or least resource consuming switch case, and all activities take their predicted duration. First, we generate a nominal schedule and find the time at which the nominal switch case is scheduled to complete, as shown in Figure 2.

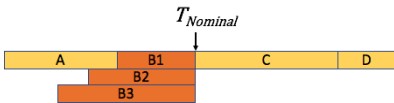

Figure 2: A, B1, C, and D are all mandatory activities in the nominal schedule. $T_{Nominal}$ is the time at which B1 is scheduled to end.

We then manipulate the execution time constraints of the more resource intensive switch cases, B2 and B3 in Figure 2, so that they are constrained to complete by $T_{Nominal}$ as shown in Equation 2. Thus, a more (time) resource consuming switch case will not use up time from any remaining lower priority mandatory activities. If an activity has more than one start time window, then we only alter the one which contains $T_{Nominal}$ and remove the others. If a prior activity ends earlier than expected during execution and frees up some time, then it may be possible to schedule a more consumptive switch case while obeying the time guard given by the altered execution time constraints.

$$T_{B_{ij\_end}} = T_{Nominal} - d_{B_i} \qquad (2)$$

Since we found that the above method was quite conservative and heavily constrained the placement of a more resource consuming switch case, we attempted a *preferred time method* to loosen the time guard. In this approach, we set the preferred time of the nominal switch case to its latest start time before generating the nominal schedule. Then, while the nominal schedule is being generated, the scheduler will try to place the nominal switch case as late as possible since the scheduler will try to place an activity as close to its preferred time as possible. As a result, $T_{Nominal}$ will likely be later than what it would be if the preferred time were not set in this way. As per Equation 2, the latest start times, $T_{B_{ij\_end}}$, of the more resource consuming switch cases may be later than what they would be using the previous method where the preferred time was not altered, thus allowing for wider start time windows for higher level switch cases. This method has some risks. If the nominal switch case was placed as late as possible, it could use up time from another mandatory activity with a tight execution window that it would not otherwise have used up if it was placed earlier, as shown in Figure 3.

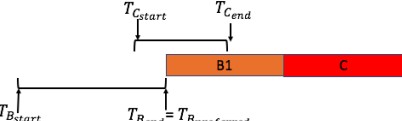

Figure 3: Scheduling B1 at its latest start time prevents C from being scheduled within its start time window.

### Guarding for Energy

**Fixed Point Minimum State of Charge Guard** The Fixed Point method attempts to ensure that scheduling a more resource consuming switch case will not cause the energy to violate the Minimum SOC while scheduling any future mandatory activities by reserving sufficient energy at a single, fixed point in time, $T_{Nominal}$ as shown in Figure 4. The guard value for the Minimum SOC is the state of charge value at $T_{Nominal}$ while constructing the nominal schedule. When attempting to schedule a more resource intensive switch case, a constraint is placed on the scheduler so that the energy cannot fall below the Minimum SOC guard value at time $T_{Nominal}$. If an activity ends early (and uses

fewer resources than expected) during execution, it may be possible to satisfy this guard while scheduling a more consumptive switch case.

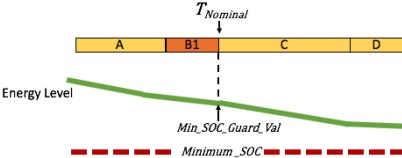

Figure 4: A, B1, C, and D, are mandatory activities in the nominal schedule. A constraint is placed so that the energy cannot dip below $Min\_SOC\_Guard\_Val$ at time $T_{Nominal}$ while trying to schedule a higher level switch case.

**Fixed Point Handover State of Charge Guard** The Fixed Point method guards for the Minimum Handover SOC by first calculating how much extra energy is left over in the nominal schedule at handover time after scheduling all activities, as shown in Figure 5.

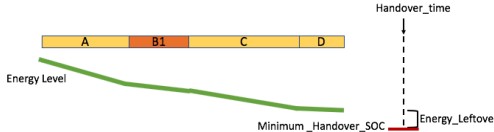

Figure 5: A, B1, C, and D, are mandatory activities in the nominal schedule. A constraint is placed so that the extra energy a higher level switch case consumes cannot exceed *Energy_Leftover*.

Then, while attempting to place a more consumptive switch case, a constraint is placed on the scheduler so that the extra energy required by the switch case does not exceed *Energy_Leftover* from the nominal schedule as in Figure 5. For example, if we have a switch group consisting of three activities, B1, B2, and B3 and $d_{B3} > d_{B2} > d_{B1}$ and each switch case consumes $e$ Watts of power, we must ensure that the following inequality holds at the time the scheduler is attempting to schedule a higher level switch case:

$$(d_{B_i} \times e_{B_i}) - (d_{B_1} \times e_{B_1}) \geq Energy\_Leftover \quad (3)$$

There may be more than one switch group in the schedule. Each time a higher level switch case is scheduled, the *Energy_Leftover* value is decreased by the extra energy required to schedule it. When the scheduler tries to place a switch case in another switch group, it will check against the updated *Energy_Leftover*.

**Sol Wide Handover State of Charge Guard** The Sol Wide handover SOC guard only schedules a more resource consumptive switch case if doing so will not cause the energy to dip below the Handover SOC at handover time. First, we use the nominal schedule to calculate how much energy is needed to schedule remaining mandatory activities.

Having a Maximum SOC constraint while calculating this value may produce an inaccurate result since any energy that would exceed the Maximum SOC would not be taken into account. So, in order to have an accurate prediction of the energy balance as activities are being scheduled, this value is calculated assuming there is no Maximum SOC constraint. 8. The Maximum SOC constraint is only removed while computing the guard offline to gain a clear understanding of the energy balance but during execution it is enforced

As shown in Figure 6, the energy needed to schedule the remaining mandatory activities is the difference between the energy level just after the nominal switch case has been scheduled, call this E1, and after all activities have been scheduled, call this energy level E2.

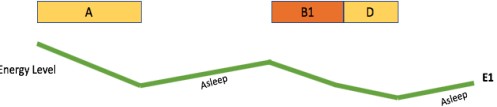

(a) E1 is the energy level of the nominal schedule with no Maximum SOC constraint after all activities up to and including the nominal switch case (A, D, B1) have been scheduled.

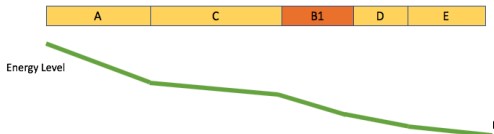

(b) E2 is the energy level of the nominal schedule with no Maximum SOC constraint after all activities in the nominal schedule have been scheduled. The activities were scheduled the following order: A, D, B1, C, E.

Figure 6: Calculating Energy Needed to Schedule Remaining Mandatory Activities.

$$Energy\_Needed = E1 - E2 \quad (4)$$

Then, a constraint is placed on the scheduler so that the energy value after a higher level switch case is scheduled must be at least:

$$Energy\_Level \geq Minimum\_Handover\_SOC \\ + Energy\_Needed \quad (5)$$

By placing this energy constraint, we hope to prevent the energy level from falling under the Minimum Handover SOC by the time all activities have been scheduled.

**Sol Wide Minimum State of Charge Guard** While we ensure that the energy will not violate the minimum Handover SOC by keeping track of the energy balance, it is possible that scheduling a longer switch case will cause the energy to fall below the Minimum SOC. To limit the chance of this happening, we run a Monte Carlo of execution offline while computing the sol wide energy guard. We use this Monte Carlo to determine if a mandatory activity was

not scheduled due to a longer switch case being scheduled earlier. If this occurs in any of the Monte Carlos of execution, then we increase the guard constraint in Equation 5. We first find the times at which each mandatory activity was scheduled to finish in the nominal schedule. Then, we run a Monte Carlo of execution with the input plan containing the guard and all switch cases. Each Monte Carlo differs in how long each activity takes to execute compared to its original predicted duration in the schedule. If a mandatory activity was not executed in any of the Monte Carlo runs and a more resource consuming switch case was executed before the time at which that mandatory activity was scheduled to complete in the nominal schedule, then we increase the Sol Wide energy guard value in Equation 5 by a fixed amount. We aim to compose a better heuristic to increase the guard value as we continue work on this subject.

## Multiple Scheduler Invocation Approach

The Multiple Scheduler Invocation (MSI) approach emulates backtracking by reinvoking the scheduler multiple times with the switch cases. MSI does not require any precomputation offline before execution as with the guards and instead reinvokes the scheduler multiple times during execution. During execution, the scheduler reschedules (e.g., when activities end early) with only the nominal switch case as shown in Figure 7a until an MSI trigger is satisfied. At this point, the scheduler is reinvoked multiple times, at most once per switch case in each switch group. In the first MSI invocation, the scheduler attempts to schedule the highest level switch case as shown in Figure 7b. If the resulting schedule does not contain all mandatory activities, then the scheduler will attempt to schedule the next highest level switch case, as in 7c, and so on. If none of the higher level switch cases can be successfully scheduled then the schedule is regenerated with the nominal switch case. If activities have ended early by the time MSI is triggered and resulted in more resources than expected, then the goal is for this approach to generate a schedule with a more consumptive switch case if it will fit (assuming nominal activity durations for any activities that have not yet executed).

There are multiple factors that must be taken into consideration when implementing MSI:

**When to Trigger MSI**   There are two options to trigger the MSI process (first invocation while trying to schedule the switch case):

1. *Time Offset.* Start MSI when the current time during execution is some fixed amount of time, $X$, from the time at which the nominal switch case is scheduled to start in the current schedule (shown in Figure 8).

2. *Switch Ready.* Start MSI when an activity has finished executing and the nominal switch case activity is the next activity scheduled to start (shown in Figure 9).

**Spacing Between MSI Invocations**   If the highest level switch case activity is not able to be scheduled in the first invocation of MSI, then the scheduler must be invoked again. We choose to reschedule as soon as possible after the most recent MSI invocation. This method risks over-consumption

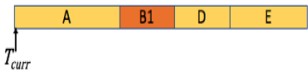

(a) MSI has not yet begun. Currently, the nominal switch case, B1, is scheduled.

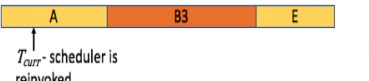

(b) MSI begins. Scheduling the highest level switch case, B3, prevents D from being scheduled. Therefore, try B2.

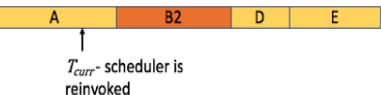

(c) B2 is successfully scheduled along with the other mandatory activities so MSI is complete.

Figure 7: Order of MSI Invocations.

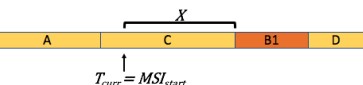

Figure 8: MSI Time Offset.

of the CPU if the scheduler is invoked too frequently. To handle this, we may need to rely on a process within the scheduler called *throttling*. Throttling places a constraint which imposes a minimum time delay between invocations, preventing the scheduler from being invoked at too high of a rate. An alternative is to reschedule at an evenly split, fixed cadence to avoid over-consumption of the CPU; we plan to explore this approach in the future.

**Switch Case Becomes Committed**   In some situations, the nominal switch case activity in the original plan may become committed before or during the MSI invocations as shown in Figure 10. An activity is *committed* if its scheduled start time is between the start and end of the commit window (Chien et al. 2000). A committed activity cannot be rescheduled and is committed to execute. If the nominal switch case remains committed, the scheduler will not be able to elevate to a higher level switch case.

There are two ways to handle this situation:

1. *Commit the activity.* Keep the nominal switch case activity committed and do not try to elevate to a higher level switch case.

2. *Veto the switch case.* Veto the nominal switch case so that it is no longer considered in the current schedule. When an activity is vetoed, it is removed from the current schedule and will be considered in a future invocation of the scheduler. Therefore, by vetoing the nominal switch case,

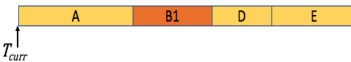

(a) B1 is the nominal switch case. Since an activity has not finished executing and B1 is not the next activity, MSI cannot begin yet.

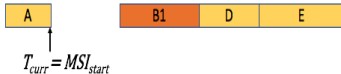

(b) Since A finished executing early, and B1 is the next activity, the MSI process can begin.

Figure 9: MSI Switch Ready.

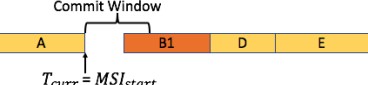

Figure 10: Switch case is committed during MSI. $T_{curr}$ is the current time during execution. $MSI_{start}$ is the time at which MSI begins. The nominal switch case, B1, is committed when MSI begins.

it will no longer be committed and the scheduler will continue the MSI invocations in an effort to elevate the switch case.

**Handling Rescheduling After MSI Completes but before the Switch Case is Committed** After MSI completes, there may be events that warrant rescheduling (e.g., an activity ending early) before the switch case is committed. When the scheduler is reinvoked to account for the event, it must know which level switch case to consider. If we successfully elevated a switch case, we choose to reschedule with that higher level switch case. Since the original schedule generated by MSI with the elevated switch case was in the past and did not undergo changes from this rescheduling, it is possible the schedule will be inconsistent and may lead to complications while scheduling later mandatory activities. An alternative we plan to explore in the future is to disable rescheduling until the switch case is committed. However, this approach would not allow the scheduler to regain time if an activity ended early and caused rescheduling.

## Empirical Analysis

In order to evaluate the performance of the above methods, we apply them to various sets of inputs comprised of activities with their constraints and compare them against each other. The inputs are derived from *sol types*. *Sol types* are currently the best available data on expected Mars 2020 rover operations (Jet Propulsion Laboratory 2017a). In order to construct a schedule and simulate plan execution, we use the *Mars 2020 surrogate scheduler* - an implementation of the same algorithm as the Mars 2020 onboard scheduler (Ra-

bideau and Benowitz 2017), but intended for a Linux workstation environment. As such, it is expected to produce the same schedules as the operational scheduler but runs much faster in a workstation environment. The surrogate scheduler is expected to assist in validating the flight scheduler implementation and also in ground operations for the mission (Chi et al. 2018).

Each sol type contains between 20 and 40 activities. Data from the Mars Science Laboratory Mission (Jet Propulsion Laboratory 2017b; Gaines et al. 2016a; 2016b) indicates that activity durations were quite conservative and completed early by around 30%. However, there is a desire by the mission to operate with a less conservative margin to increase productivity. In our model to determine activity execution durations, we choose from a normal distribution where the mean is 90% of the predicted, nominal activity duration. The standard deviation is set so that 10 % of activity execution durations will be greater than the nominal duration. For our analysis, if an activity's execution duration chosen from the distribution is longer than its nominal duration, then the execution duration is set to be the nominal duration to avoid many complications which result from activities running long (e.g., an activity may not be scheduled solely because another activity ran late). Detailed discussion of this is the subject of another paper. We do not explicitly change other activity resources such as energy and data volume since they are generally modeled as rates and changing activity durations implicitly changes energy and data volume as well.

We create 10 variants derived from each of 8 sol types by adding one switch group to each set of inputs for a total of 80 variants. The switch group contains three switch cases, $A_{nominal}$, $A_{2x}$, and $A_{4x}$ where $d_{A_{4x}} = 4 \times d_{A_{nominal}}$ and $d_{A_{2x}} = 2 \times d_{A_{nominal}}$.

In order to evaluate the effectiveness of each method, we have developed a scoring method based on how many and what type of activities are able to be scheduled successfully. The score is such that the value of any single mandatory activity being scheduled is much greater than that of any combination of switch cases (at most one activity from each switch group can be scheduled).

Each mandatory activity that is successfully scheduled, including whichever switch case activity is scheduled, contributes one point to the *mandatory score*. A successfully scheduled switch case that is 2 times as long as the original activity contributes $1/2$ to the *switch group score*. A successfully scheduled switch case that is 4 times as long as the original, nominal switch case contributes 1 to the switch group score. If only the nominal switch case is able to be scheduled, it does not contribute to the switch group score at all. There is only one switch group in each variant, so the maximum switch group score for a variant is 1. Since scheduling a mandatory activity is of much higher importance than scheduling any number of higher level switch case, the mandatory activity score is weighted at a much larger value then the switch group score. In the following empirical results, we average the mandatory and switch groups scores over 20 Monte Carlo runs of execution for each variant.

We compare the different methods to schedule switch cases over varying incoming state of charge values (how much energy exists at the start) and determine which methods result in 1) scheduling all mandatory activities and 2) the highest switch group scores. The upper bound for the theoretical maximum switch group score is given by an *omniscient scheduler*- a scheduler which has prior knowledge of the execution duration for each activity. Thus, this scheduler is aware of the amount of resources that will be available to schedule higher level switch cases given how long activities take to execute compared to their predicted, nominal duration. The input activity durations fed to this omniscient scheduler are the actual execution durations. We run the omniscient scheduler at most once per switch case. First, we try to schedule with only the highest level switch case and if that fails to schedule all mandatory activities, then we try with the next level switch case, and so on.

First, we determine which methods are able to successfully schedule all mandatory activities, indicated by the Maximum Mandatory Score in Figure 11. Since scheduling a mandatory activity is worth much more than scheduling any number of higher level switch cases, we only compare switch group scores between methods that successfully schedule all mandatory activities.

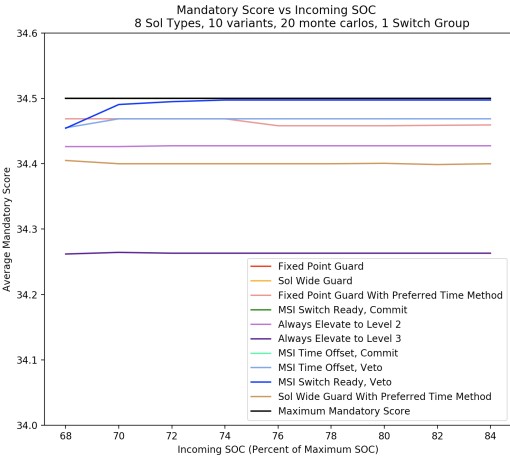

Figure 11: Mandatory score vs Incoming SOC for various Methods to Schedule Switch Cases

In order to evaluate the ability of each method to schedule all mandatory activities, we also compare against two other methods, one which always elevates to the highest level switch case while the other always elevates to the medium level switch case. We see in Figure 11 that always elevating to the highest (3rd) level performs the worst and drops approximately 0.25 mandatory activities per sol, or 1 activity per 4 sols on average while always elevating to the second highest level drops close to 0.07 mandatory activities per sol, or 1 activity per 14 sols on average. For comparison, the study described in (Gaines et al. 2016a) showed that approximately 1 mandatory activity was dropped every 90 sols, indicating that both of these heuristics perform poorly.

We found that using preferred time to guard against time

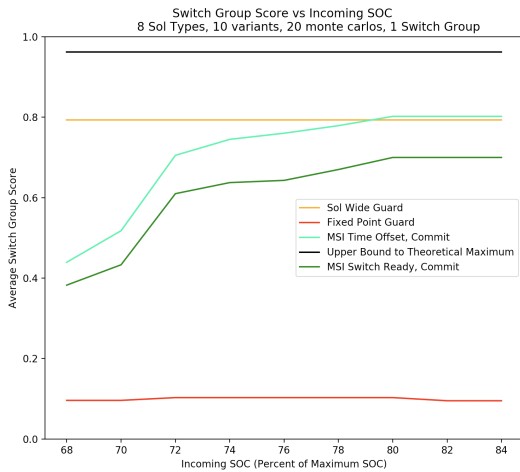

Figure 12: Switch Group Score vs Incoming SOC for Methods which Schedule all Mandatory Activities

caused mandatory activities to drop for both the fixed point and sol wide guard (for the reason described in the Guarding for Time section) while using the original method to guard against time did not. We see in Figure 11 that the preferred time method with the fixed point guard drops on average about 0.04 mandatory activities per sol, or 1 activity every 25 sols while the sol wide guard drops on average about 0.1 mandatory activities per sol, or 1 activity every 10 sols. We also see that occasionally fewer mandatory activities are scheduled with a higher incoming SOC. Since using preferred time does not properly ensure that all remaining activities will be able to be scheduled, a higher incoming SOC can allow a higher level switch case to be scheduled, preventing future mandatory activities from being scheduled.

The MSI approaches which veto to handle the situation where the nominal switch case becomes committed before or during MSI drop mandatory activities. Whenever an activity is vetoed, there is always the risk that it will not be able to be scheduled in a future invocation, more so if the sol type is very tightly time constrained, which is especially true for one of our sol types. Thus, vetoing the nominal switch case can result in dropping the activity, accounting for this method's inability to schedule all mandatory activities. The MSI methods that keep the nominal switch case committed and do not try to elevate to a higher level switch case successfully schedule all mandatory activities, as do the guard methods.

We see that the Fixed Point guard, Sol Wide guard, and two of the MSI approaches are able to successfully schedule all mandatory activities. As shown in Figure 12, the Sol Wide guard and MSI approach using the options Time Offset and Commit result in the highest switch group scores closest to the upper bound for the theoretical maximum. Both MSI approaches have increasing switch group scores with increasing incoming SOC since a higher incoming energy will result in more energy to schedule a consumptive switch case during MSI. The less time there is to complete all MSI

invocations, the more likely it is for the nominal switch case to become committed. Since we give up trying to elevate switch cases and keep the switch case committed if this occurs, fewer switch cases will be elevated. Because our time offset value, $X$, in Figure 8 is quite large (15 minutes), this situation is more likely to occur using the Switch Ready approach to choose when to start MSI, explaining why using Switch Ready results in a lower switch score than Time Offset.

The Fixed Point guard results in a significantly lower switch case score because it checks against a state of charge constraint at a particular time regardless of what occurs during execution. Even if a switch case is being attempted to be scheduled at a completely different time than $T_{Nominal}$ in Figure 2, (e.g., because prior activities ended early), the guard constraint will still be enforced at that particular time. Since we simulate activities ending early, more activities will likely complete by $T_{Nominal}$, causing the energy level to fall under the Minimum SOC Guard value. Unlike the Fixed Point guard, since the the Sol Wide guard checks if there is sufficient energy to schedule a higher level switch case at the time the scheduler is attempting to schedule it, not at a set time, it is better able to consider resources regained from an activity ending early.

We also see that using the Fixed Point guard begins to result in a lower switch group score with higher incoming SOC levels after the incoming SOC is 80% of the Maximum SOC. Energy is more likely to reach the Maximum SOC constraint with a higher incoming SOC. The energy gained by an activity taking less time than predicted will not be able to be used if the resulting energy level would exceed the Maximum SOC. If this occurs, then since the extra energy cannot be used, the energy level may dip below the guard value in Figure 4 at time $T_{Nominal}$ while trying to schedule a higher level switch case even if an activity ended sufficiently early, as shown in Figure 13.

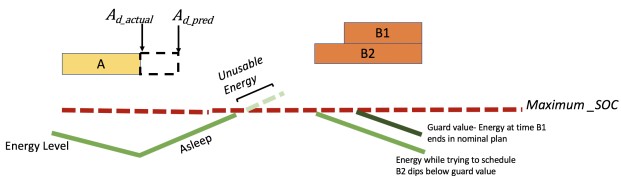

Figure 13: Fixed Point Guard Schedules Fewer Mandatory Activities with Higher Incoming SOC

## Related Work

Just-In-Case Scheduling (Drummond, Bresina, and Swanson 1994) uses a nominal schedule to determine areas where breaks in the schedule are most likely to occur and produces a branching (tree) schedule to cover execution contingencies. Our approaches all (re) schedule on the fly although the guard methods can be vewied as forcing schedule branches based on time and resource availability.

Kellenbrink and Helber (Kellenbrink and Helber 2015) solve RCPSP (resource-constrained project scheduling

problem) where all activities that must be scheduled are not known in advance and the scheduler must decide whether or not to perform certain activities of varying resource consumption. Similarly, our scheduler does not know which of the switch cases to schedule in advance, using runtime resource information to drive (re) scheduling.

Integrated planning and scheduling can also be considered scheduling disjuncts (chosen based on prevailing conditions (e.g., (Barták 2000))) but these methods typically search whereas we are too computationally limited to search.

## Discussion and Future Work

There are many areas for future work. Currently the time guard heavily limits the placement of activities. As we saw, using preferred time to address this issue resulted in dropping mandatory activities. Ideally analysis of start time windows and dependencies could determine where an activity could be placed without blocking other mandatory activities.

Additionally, in computing the guard for Minimum SOC using the Sol Wide Guard, instead of increasing the guard value by a predetermined fixed amount which could result in over-conservatism, binary search via Monte Carlo analysis could more precisely determine the guard amount.

Currently we consider only a single switch group per plan, the Mars 2020 rover mission desires support for multiple switch groups in the input instead. Additional work is needed to extend to multiple switch groups.

Further exploration of all of the MSI variants is needed. Study of starting MSI invocations if an activity ends early by at least some amount and the switch case is the next activity is planned. We would like to analyze the effects of evenly spacing the MSI invocations in order to avoid relying on throttling and we would like to try disabling rescheduling after MSI is complete until the switch case has been committed and understand if this results in major drawbacks.

We have studied the effects of time and energy on switch cases, and we would like to extend these approaches and analysis to data volume.

## Conclusion

We have presented several algorithms to allow a very computationally limited, non-backtracking scheduler to consider a schedule containing required, or mandatory, activities and sets of activities called switch groups where each activity in such sets differs only by its resource consumption. These algorithms strive to schedule the most preferred, which happens to be the most consumptive, activity possible in the set without dropping any other mandatory activity. First, we discuss two guard methods which use different approaches to reserve enough resources to schedule remaining mandatory activities. We then discuss a third algorithm, MSI, which emulates backtracking by reinvoking the scheduler at most once per level of switch case. We present empirical analysis using input sets of activities derived from data on expected planetary rover operations to show the effects of using each of these methods. These implementations and empirical evaluation are currently being evaluated in the context of the Mars 2020 onboard scheduler.

## Acknowledgments

This work was performed at the Jet Propulsion Laboratory, California Institute of Technology, under a contract with the National Aeronautics and Space Administration.

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
