# OpenReview forum: "Enabling Limited Resource-Bounded Disjunction in Scheduling"
_icaps-conference.org/ICAPS/2019/Workshop/SPARK — SPARK 2019_

### Official Review · AnonReviewer2 · 2019-04-26
**Interesting work, though apparently still preliminary. Very interesting application domain. Very suitable material for SPARK.**

**Rating:** 3
**Confidence:** 2

**Review:**

This paper analyzes three approaches to tackle the (re)scheduling problem of typical plans to be executed onboard a planetary rover. The main challenge to be solved in this work is the limited amount of computational resources to execute the scheduling process, as such process is supposed to occur on board. The presented analysis is therefore a study tailored on the actual computational capabilities of a rover-embedded scheduler, whose performance is necessarily limited.

This work is a crisp example of scheduling technology applied to a real-world applicative context even though, as far as the reviewer understands, the study revolves around possible techniques tailored for the real Mars 2020 onboard scheduler but implemented on only a surrogate of the same scheduler. The analysis is therefore intended to provide just a validation of the proposed scheduling techniques.

This work is certainly of great interest for the SPARK audience, and may generate a lively discussion ay the workshop. Also, the model of the applicative domain may potentially represent a very close approximation of the real-world scheduling instances that could be met by the real rover.

Some perplexities arise only in the fact that the experimental conditions used in this work are still relatively simple, as the authors adopt some simplifying hypotheses (for instance, all guards are computed disregarding the data volume and considering time and energy only; moreover, the energy issues are reduced to temporal considerations only, as the consumed energy is directly proportional to the activity durations, therefore any constraint on energy is directly translated to a temporal constraint). The reviewer understands that this is an ongoing work, and that the obtained results are still preliminary. Many points in the analysis are still open, and the impression is that a significant amount of work is necessary to fill all the gaps mentioned by the authors. But the level on maturity of the presented material is suitable for a workshop.

Below, some comments that may help the authors improve the contents/clarity of the paper.

- Some of the subsections describing the various cases (e.g., Fixed Point Vs. Sol Wide) are awkwardly ordered, see for instance the "Guarding for Energy" section.

- Section "Sol Wide Minimum State of Charge Guard": the authors claim that they are working on a better heuristic w.r.t. the one currently implemented. Truly, the current heuristic seems very costly to the reviewer, and hardly suitable for hard real-time utilization. Any comment?

- Hoe would the approach fit when deadlines on activity end times are added to the scenario? (e.g., plans having to end before martian sunset). The reviewer understands that at the time being, unexpectedly longer durations are cut to the nominal durations, hence only "shorter than expected" durations are taken into account.

- In the current scenario, are minimum battery requirements to keep all instrument "warm" in order to survive the freezing martian, satisfied?

- To the best of the reviewer's understanding, one of the most important issues in this work is the decision about how often to reschedule, as well as the issue of managing reschedulings in the presence of committed activities. Especially related to the second point, this raises the general issue of system's latency. What are the rescheduling time constants associated to the proposed approach, relatively to the problem instances selected for the empirical analysis?

- One last comment on the empirical analysis. The section is not very clear. It is sometimes hard to follow the textual description of the results based on the figures, and to check their correspondence to what is depicted in the same figures. For instance, some numbers are mentioned in the textual description that the figures do not show.

Despite the approach analyzed in this work raises some perplexities on the real hard real-time applicability of the current implementation, the reviewer believes that the presented material is sufficiently interested to be accepted.

---

### Official Review · AnonReviewer1 · 2019-05-01
**The scheduing scenario addressed by this paper needs clarification**

**Rating:** 4
**Confidence:** 2

**Review:**

This paper describes alternate approaches to scheduling activities in an environment with scarce computational resources such that a backtracking search is not possible.  Three approaches are taken to ensuring that high priority tasks are not missed.  The first two approaches provide guards on a nominal pre-defined schedule and the third approach introduces a limited form of backtracking by making multiple schedule runs.

The paper presents the work in terms of scheduling activities on board the planned 2020 Mars rover. As such the results are interesting to the SPARK community.

The paper is confusing as to whether its goal is to create a schedule versus the ability to adjust an existing schedule during execution to take advantage of shorter than modeled activity durations.   The paper introduces the concept of a switch set which is priority ordered set of activities out of which one must be scheduled with a preference to schedule the highest priority activity that still allows all the other mandatory activities to schedule.  Most of the paper involves scheduling switch sets.  While switch sets are interesting,  I am not sure how the problem is solved without switch sets.   Scheduling is an intractable problem.  How does the approach get good schedules with no backtracking?  This is especially concerning given that the evaluation problems only have a single switch set.

The authors need to clarify the scenario in which the planner will operate and how the new approaches impact this scenario.

Based on the example given the approach guarding for time is not sound and can result in schedules that do not schedule all mandatory activities.

---

### Decision · Program_Chairs · 2019-05-08
**Acceptance Decision**

**Decision:**

Accept

**Comment:**

Interesting and relevant for SPARK.